# Microplastics in Agricultural Crops and Their Possible Impact on Farmers’ Health: A Review

**DOI:** 10.3390/ijerph22010045

**Published:** 2024-12-31

**Authors:** Eva Masciarelli, Laura Casorri, Marco Di Luigi, Claudio Beni, Massimiliano Valentini, Erica Costantini, Lisa Aielli, Marcella Reale

**Affiliations:** 1Department of Technological Innovations and Safety of Plants, Products and Anthropic Settlements, National Institute for Insurance Against Accidents at Work, Via R. Ferruzzi, 38/40, 00143 Rome, Italy; e.masciarelli@inail.it (E.M.); l.casorri@inail.it (L.C.); 2Department of Occupational and Environmental Medicine, Epidemiology and Hygiene, National Institute for Insurance Against Accidents at Work, Via di Fontana Candida, 1, Monte Porzio Catone, 00078 Rome, Italy; 3Research Centre for Engineering and Agro-Food Processing, Council for Agricultural Research and Economics, Via della Pascolare, 16, Monterotondo, 00015 Rome, Italy; claudio.beni@crea.gov.it; 4Research Centre for Food and Nutrition, Council for Agricultural Research and Economics, Via Ardeatina, 546, 00178 Rome, Italy; massimiliano.valentini@crea.gov.it; 5Department Innovative Technologies in Medicine and Dentistry, University “G. d’Annunzio”, Via dei Vestini, 66100 Chieti, Italy; erica.costantini@unich.it (E.C.); lisa.aielli@unich.it (L.A.); mreale@unich.it (M.R.)

**Keywords:** microplastics, impact on agricultural, crops, environmental safety, occupational risk, human health, agricultural workers safety

## Abstract

The indiscriminate use of plastic products and their inappropriate management and disposal contribute to the increasing presence and accumulation of this material in all environmental zones. The chemical properties of plastics and their resistance to natural degradation lead over time to the production of microplastics (MPs) and nanoplastics, which are dispersed in soil, water, and air and can be absorbed by plants, including those grown for food. In agriculture, MPs can come from many sources (mulch film, tractor tires, compost, fertilizers, and pesticides). The possible effects of this type of pollution on living organisms, especially humans, increase the need to carry out studies to assess occupational exposure in agriculture. It would also be desirable to promote alternative materials to plastic and sustainable agronomic practices to protect the safety and health of agricultural workers.

## 1. Introduction

Since its invention around the beginning of the 1900s, plastic has shown wide potential for use [1]. In just a few decades, plastic has invaded our lives (packaging, synthetic textile fibers, disposable plastic products, components, and structural parts of vehicles), and in recent years, its exponential growth production has broadly exceeded that of any other material [2]. In everyday use, plastic advantages such as versatility, practicality, strength, durability, and low cost are recognized worldwide [3]. For this reason, plastic objects are preferred to those made of similar, less resistant, and perishable materials. However, the environmental problems of plastic are precisely due to its endurance. Plastic objects take several hundred years to degrade, but often they are used only once or for a short period. As a result, plastic waste accumulates in all environmental matrices and contaminates them [4,5], and their debris are found all over the whole planet [6,7,8]. Since the 1950s, the worldwide production of plastics reached 400.3 million tons in 2022, marking an increase of about 1.6 percent from the previous year [9]. By 2050, around 12,000 megatons of plastic will be dispersed in the environment, mainly in the oceans and on agricultural soil. In 2014, the first United Nations Environment Assembly defined marine pollution due to plastic waste as one of the 10 most urgent environmental problems to be solved. Only 9% of plastic is recycled, and the rest is incinerated, buried, or thrown into the environment [10]. The growing environmental plastic presence and its global distribution make pollution from polymeric materials one of the most prominent environmental problems, which also concerns natural oases and remote areas such as mountain tops and ocean depths [7,11,12,13]. Due to their limited recovery and great resistance to degradation, plastic residues are increasingly accumulating in the environment, where they can persist for long periods and disintegrate into small plastic particles [14]. This phenomenon is known as “white pollution” [15]. Everywhere, from aquatic ecosystems to land, from crowded beaches to remote Arctic ice, microscopic plastic debris are found now. Thompson et al. [16] introduced the term microplastic (MP) to indicate plastic debris. A more detailed definition of MPs proposed by Frias and Nash [17] defines them as synthetic solid particles or polymer matrices, with regular or irregular shapes and with sizes ranging from 1 μm to 5 mm. MPs are a topic of global environmental concern as a class of emerging and quasi-permanent contaminants [18,19,20,21,22,23,24]. MPs are insoluble in water, accumulate, and interact with the environment. These pollutants are potentially among the most dangerous for the ecosystems’ future because they can modify the physical and chemical parameters of soil and vegetables. On land, contamination by MPs has been found to be 4 to 23 times greater than in the ocean [23,25]. MPs have also been found in seafood, fish, and cetacean tissues, and even in salt. Scientists have become concerned about understanding if and how MPs could transfer from the environment to the food chain and what their potential impact on human health is because these less than 5 mm tiny plastic fragments are ubiquitous. The MPs presence has been detected in soil, food, bottled water, and even in the organism and human body. MPs have also been found in the placenta and human feces. Toxicity tests have shown that MPs can negatively affect the growth, reproduction, and endocrine systems of human organisms [26,27]. Furthermore, plastic translocation can occur between organisms through animal predation, and plastic can migrate between tissues and organs through an organism’s circulatory system [28,29,30]. MPs are crumbled by external environmental factors into smaller particles, even on the nanoscale, and finally could be degraded into CO_2_, H_2_O, and methane [31]. One of the most effective abiotic MPs degradation processes is photodegradation. Airborne, floating on the water, and spread on the soil, MPs, when exposed to sunlight, gradually degrade under the ultraviolet light and atmospheric oxygen actions [32]. MPs are classified into primary and secondary. The primary MPs are plastic particles intentionally manufactured to be added to some products (e.g., abrasive granules in cosmetics) [33]. In contrast, the secondary ones are produced by the larger plastic products’ disintegration produced by natural (atmospheric action such as light, ice, etc.) or anthropogenic actions (mechanical abrasion, synthetic fabrics washing, etc.) [24,30,34,35,36]. The danger of these particles is also due to their ability to act as carriers for both microorganisms (*Aeromonas*, *Rhodococcus*, *Pseudomonas*, *Enterobacter*, *Halomonas*, *Mycobacterium*, *Photobacterium*, and *Shigella*, and fungi) and organic and inorganic contaminants [14,18,37,38], potentially increasing the risk extent to living organisms of harm [39]. In addition, the environmental persistence of polymeric materials not only makes plastic pollution a perennial problem but also exposes the particles to continuous aging processes (mechanical deterioration, photooxidation, chemical alteration, and biological decomposition) [40,41,42]. These processes could increase their toxicity by modifying their structure, size, morphology, and porosity [43]. One of the greater users of plastic is the agricultural sector. There is a lot of plastic waste in agricultural ecosystems derived from slow-release fertilizer polymers (in Europe up to around 8000 tons) [44] and from mulch and packaging residues. Agroecosystems are considered the main source of MPs pollution in terrestrial environments [45]. Indeed, in these ecosystems, primary MPs come from the use of wastewater and sludge for soil fertilization. Secondary MPs can derive from the degradation of plastic sheets, used for mulching or greenhouse covering, due to ultraviolet (UV) radiation, high temperatures, mechanical abrasion, and biological degradation [25,46,47,48,49,50]. Airborne MPs contribute to air pollution. The atmospheric fallout MPs dominant polymer components are polypropylene (PP), polyethylene (PE), polystyrene (PS), and polyethylene terephthalate (PET) [15]. Some studies describe the impacts and possible MPs effects on the environment and organisms. However, due to their complexity and the lack of accurate determination methods, the systematic impact of MPs pollution on trophic chains (soils, plants, and animals) is still unclear [30,51,52,53,54,55,56,57,58,59]. In this context, the terrestrial plants’ response, primary producers that contribute to regulating the global climate, to MPs exposures should be better studied [60]. Plant exposure to MPs mainly through water and soil occurs [53,58]. A soil kilogram can hold over 40,000 microplastic particles, mainly deriving from the larger parts degradation, such as fibers (92%) or bigger fragments (4.1%) [61]. Many studies show that MPs harm plants, mainly by altering the soil structure and its microbiological characteristics and also through direct contact with the root surface. Sediment-rooted aquatic macrophytes exposed to MPs exhibit reduced growth rate and cell viability [62,63], leaf changes, and effects due to ecotoxicity and inducing oxidative damage [64]. Land plants exposed to MPs show reduced shoot and root biomass, reduced germination rate, genotoxic and oxidative damage, photosynthesis reduction, and changes in elemental composition and metabolic profiles [50,57,65,66,67,68,69,70].

According to a study by the Pacific Northwest National Laboratory (PNNL) and Washington State University (WSU), MPs may be able to enter plant cells. However, MPs accumulate on root tips (not taproots), suggesting that the plants could be used as phytopurifiers [71]. Based on these studies, it would be interesting to investigate how vascular plants and MPs interact and what the MPs’ phytotoxic effects on vascular plants are [30]. This study examines the distribution of MPs in the environmental matrices of air, water, and soil. It then considers the subsequent impact of these factors on plants of agricultural interest, grown for human and animal consumption, and their potential implications for the health of consumers and agricultural workers.

## 2. Current Legislation

In response to the issues related to MPs, from 17 October 2023, with Regulation 2023/2055/EU [72], amendments were made to Regulation 1907/2006/EC on the Registration, Evaluation, Authorization, and Restriction of Chemicals and Mixtures (REACH) [73] to include microplastics among the substances subject to restrictions. This regulation provides a comprehensive definition of microplastics that includes all particles (organic, insoluble, and resistant to degradation) of synthetic polymers smaller than five mm. The measure modifies Annex XVII to Regulation 1907/2006/EC [73] (products subject to a ban on manufacturing, use, and placing on the market). The restriction on microplastics will include granular infill material used for artificial sports surfaces, cosmetics, detergents, fabric softeners, glitter, fertilizers, plant protection products, toys, medicines, and medical devices. For controlled-release fertilizers, a transitional period of five years has been established to allow manufacturers to reformulate their products to achieve adequate degradability in the environment.

For plant protection products (European Parliament and Council Regulation No 1107/2009/EC [74]), seeds treated with them, and biocidal products (European Parliament and Council Regulation No 528/2012/EU [75]), a transitional period of eight years to reformulate the products concerned was granted to industries to obtain the authorization to place them on the market, maintaining the benefits of the encapsulation technology during the transitional period. For seeds coated with dyes or lubricants, or other products that are not or do not contain plant protection products, a transitional period of five years has been established [72].

## 3. MPs Sources

In 2021, packaging and building and construction applications were the two largest plastic-using sectors in the world. In fact, of the estimated total plastic production of 390.7 Mt in 2021, packaging, with about 44%, is the largest user sector, followed by building and construction (18%), automotive (8%), electrical and electronic (7%), such as home, leisure, and sports, and, at 4%, agriculture, livestock, and gardening. Other uses together reach about 12%. Polymers not used in the production of plastic parts and products (e.g., textiles, adhesives, sealants, coatings, etc.) were not considered [76].

In agricultural soils, the distribution of origin of microplastics is reported in Figure 1.

Plastics are polymeric macromolecular chain materials of different lengths composed of monomers. The most used plastics are polyethylene (PE), polypropylene (PP), polystyrene (PS), polyethylene terephthalate (PET), and polyvinyl chloride (PVC). MP particles can be divided into primary, produced by industry for various purposes (pellets used to make plastic products, abrasive microspheres, or personal care products), and secondary, which are generated by materials disintegration, abrasion, or waste released into the environment [78].

Plastics can be non-degradable, easily degradable, or biodegradable, or with controlled degradation. In the agricultural environment, plastic degradation is determined by the following:Ambiental factors (UV radiation, temperature, humidity, atmospheric agents, and meteorological phenomena);Chemicals used in cultivation;Material composition;Additives used in the formulation);Habitat (biodegradation by bacteria, fungi, and lichens; action of insects, birds, and rodents).

All these processes lead to the production of MPs (<5 μm) and nanoplastics (NPs, <0.1 μm) that can persist in the environment [79,80]. Plastics have very high environmental persistence and longevity because they are designed to last. The degradation times of some polymers are much longer than can be simulated in laboratory or field experiments, so their lifetime estimates must be based on extrapolations with considerable uncertainties. The fact that MPs are widespread in the environment, although plastics production began only 70 years ago, implies that the formation of MPs is quite rapid, with, at most, a timescale of decades [81]. MPs have a large specific surface area and high hydrophobicity and thus can easily adsorb various types of environmental pollutants (polychlorinated biphenyls (PCBs), polycyclic aromatic hydrocarbons (PAHs), organochlorine pesticides (OPCs), antibiotics, and heavy metals [82]. The toxic effects of the different components are combined and therefore more difficult to analyze [15,32,83,84].

## 4. Most Common Plastics in Agriculture

Thanks to their advantages and economic savings provided by plastic use in agriculture, they are now widespread despite the problem of produced waste and its disposal [77,85]. In agriculture, the main polymers used include polyolefins (polyethylene PE, polypropylene PP, ethylene-vinyl-acetate EVA), polyvinyl chloride (PVC), polycarbonate (PC), polymethyl methacrylate (PMMA), and glass fiber-reinforced plastic (GRP) [77,86]. The most common additives used to stabilize these materials are HALS (hindered amine light stabilizers), specific colorants, UV absorbers, UV stabilizers, antioxidants, and thermal stabilizers [85] (Table 1 and Table 2).

## 5. The Cycle of MPs in the Environment (Air, Soil, Water, Plants, Microorganisms, Organisms, the Food Chain, and Humans)

MPs move through the environment in a continuous cycle. They are deposited on the soil surface from the ocean, where they come into contact with plants. From the soil surface, they can flow back into the oceans by wind and rain (Figure 2).

### 5.1. Airborne Microplastics

It has been found that the amount of airborne MPs depends on anthropic activities (industrial, domestic, and agricultural), population density, and the level of industrialization of the land [7,19,88,89]. The most common airborne MPs are PE, PP, polyester (PES), PET [90], and nylon in the form of films and fibers.

Fibrous MPs are generally produced from fabrics during mechanical processes [91], while films are produced by the degradation of plastic bags. Main sources of airborne MPs include agricultural practices (plastic mulches and synthetic fertilizers), sewage sludge, and landfill leachate [92]. The MP particles’ dynamic cycling (aquatic, terrestrial, and airborne environments) has a significant effect on their concentrations in the air [93]. Interestingly, airborne MPs can be transported to remote areas and can remain in the air for an hour to 6.5 days [94], a time sufficient for transcontinental transport. The lowest density particles can be transported from urban areas and deposited in forest areas, and according to Boucher and Friot [95], terrestrial plastic is transferred by wind and finally falls into the ocean. Their deposition is a source of soil and water pollution (source-path-sink connectivity networks) [96]. The distribution patterns of MPs movements in the air are not yet clear. Factors such as the pollution concentration gradient, wind speed and direction, air temperature, and humidity influence the dispersion and deposition of MPs [24]. The particle sizes from films are generally larger because they come from primary sources that do not undergo secondary decomposition in the air [97]. Fate and dispersion of MPs in the airborne environment depend on the characteristics of the particles (density, length, and diameter) and environmental factors (wind and precipitation). Zhang et al. [98] observed that the characteristics of MP particles (sources and aerodynamics) can influence the pollutants’ presence and transport in the atmosphere. Guo et al. [99] studied the MPs surface adsorption mechanism and found that the oxygen-containing functional groups in polyphenylene ethers act as hydrogen bond acceptors interacting with other molecules, making MPs more susceptible to the adsorption of pollutants and microorganisms.

There are still large gaps to be filled in our knowledge about the environmental behaviors of airborne MPs, such as size, atmospheric transport, fate, and interaction between urban and ocean environments [100,101]. To better understand their behavior and the potential risks to human health, it is important to define the sources, flows, and deposition patterns of airborne MPs with greater precision. A recent study conducted in Sri Lanka, both in indoor and outdoor environments, indicates that people are probably exposed to thousands of airborne MPs every year [102]. Cox et al. [103] established that airborne MPs represent a significant source of individual daily exposure. MPs could enter the human body through inhalation and ingestion. Due to both their large specific area surfaces and great adsorption capacity, MPs often carry adsorbed contaminants (heavy metals, toxic chemicals, and microorganisms) [104,105].

Airborne MPs can interact with the ecosystem and influence it [106]. The atmospheric particulate matter has many common properties with them [107] and similarly can be captured by terrestrial plants [108] and accumulated within their leaves [109].

According to Mohan et al. [110], in terrestrial ecosystems, MPs captured by leaves through deposition or runoff from precipitation become a source of MPs for the soil. The MPs lipophilic surface properties make them chemically similar to the waxy surface of many leaves, favoring their adhesion, and some particles with particular shapes, such as microfibers, can easily intertwine with trichomes.

Significant concentrations of MPs can accumulate on leaf surfaces. They can block sunlight, decrease photosynthesis, influence the chlorophyll A concentrations, the carbon and nitrogen cycles, and the leafy ecosystem [111], interfering with plant growth, such as in freshwater algae studied by Wu et al. [112]. Eichert et al. [113] showed leaf stomata can work as a gateway for polystyrene particles between 43 nm and 1.1 μm. These can induce alterations in the mesophyll apoplast. MPs adhered to leaf surfaces can also operate as an adsorbent layer for additional environmental contaminants (volatile organic compounds or heavy metals) harmful to plant health. Edible plants grown in contaminated areas constitute potential human and animal exposure routes. Larger MPs can be removed from fruit and vegetable surfaces by washing, while smaller MPs (<10 μm), incorporated in the cuticles of plants, are resistant to removal [114]. Evangeliou et al. [115] reported that pigmented MPs can absorb more solar radiation in the visible spectrum than they scatter, contributing to atmospheric warming. In urban areas with relatively higher concentrations of airborne MPs, these particles are likely to influence atmospheric temperature, which could have a significant impact on plant growth, photosynthesis, and biochemical reactions [116,117]. The direct adverse effects of exposure to MPs are plant growth and seed germination inhibition.

Once on the ground MPs can alter the soil’s physical and chemical properties, exerting ecotoxicological effects on the organisms present in it; the problem can be worsened by the other pollutants adsorbed on them [24,118]. MPs accumulated near the roots could be absorbed by the plant [118,119], a phenomenon particularly important in edible plants [24,79,120,121].

### 5.2. Microplastics in Soil and Water

MPs, as persistent emerging contaminants in terrestrial ecosystems, directly or indirectly can affect all ecosystem components and consequently alter their functions [67,122]. Rillig [123] has been the first to propose that the accumulation of MPs in soil could potentially cause adverse effects on its properties and biodiversity, as subsequently confirmed by numerous studies [56,80,118,124,125]. MPs can alter the physical (structure, porosity, density, and water retention capacity), chemical (pH, organic matter, nutrients, and contaminants), and microbiological characteristics of the soil in the rhizosphere [118] and affect its fauna (earthworms, nematodes, etc.) [126,127]. MPs accumulation through adsorption and complexation depends on chemical soil characteristics (pH, ionic strength, cations) and can directly or indirectly influence particle fate and transport [128,129]. Jiang et al. [130] found that the type of land use can influence the concentration of microplastics, while soil properties (pH, dissolved organic matter content, and total iron content) did not appear to be the key factor influencing the abundance of MPs. Zhao et al. [131] reported that microplastics increase pH and slow down soil microbial activities. Microplastics also affect the activities of some soil enzymes (dehydrogenase, acid phosphatase, urease, peroxidase, and protease) [52]. The biota activities and soil tillage influence MPs deposition and movement in soil. The main reason for downward movement are biopores created by biota or soil cracking [132,133,134], while the horizontal movement is caused by ploughing.

The MPs’ physical properties (size, shape, and hydrophobicity) can influence their transport within the soil [132,133], while aggregation processes influence their distribution [50]. Several studies have evaluated the potential for contamination of soil by MPs.

For example, Fuller and Gautam [135] reported that the MPs concentration in an industrial area surface soil can be between 0.03 and 6.7%, while in soils from natural reserves, almost free from human activities, concentrations of up to 0.002% were detected [136].

Many human activities can contribute to MP soil contamination, including the following [25,48,79,137]:Mulching plastic sheeting fragmentation;Atmospheric or aerial deposition (from uncovered or poorly managed landfills, municipal waste);Agricultural land irrigation using contaminated water or road runoff (tires abrasion);Biosolids application (wastewater purification by-products);Plastic-coated fertilizers use;Municipal waste compost fertilization;Atmospheric precipitation.

Biosolids and plastic mulch films are the major soil plastic contaminants. Biosolids can retain and accumulate up to 99% of plastic particles after repeated or long-term application of treated sludge [123]. Fertilizer microcapsules, PVC pipe use, and mulching with plastic sheets [138] could potentially be the major sources of MPs in agricultural soils [139,140]. Often, intentionally or unintentionally, plastic sheets or parts of them are left in agricultural lands, where they can be degraded into microparticles [139]. Their improper disposal can lead to the accumulation of MPs in agricultural soils, representing a considerable threat to terrestrial fauna and food security [23,141]. In 2020, a study by Huang et al. [23] hypothesized that MPs accumulate in agricultural soils due to long-term mulching. studies on plastic residues in mulched soils were brought across China to test this hypothesis. The results of this study confirmed mulching caused the MPs contamination. Zhu et al. [142] reported that plastic films used for mulching are usually thin (about 10–30 μm), thus difficult to remove from the field and recycle, and can slowly fragment into smaller particles due to soil tillage, UV radiation, water, and wind [143]. Plastic polymers are also used in agriculture to produce microparticle coatings of fertilizers and plant protection products to control their release [72]. PCF, polymer-coated fertilizers are designed as soluble nutrient cores and thermoset polymer coatings to release nutrients in a progressively controlled manner [144]. The contribution of their residues to MPs pollution in agroecosystems has long been overlooked [145,146,147].

Katsumi et al. [148] found high MP levels in Japanese rice fields from PCF microcapsules, mainly constituted by non-biodegradable polymers such as polyethylene and polyurethane, smaller in size than the mulching sheets debris. Their removal is time-consuming and expensive, and they therefore accumulate in large quantities in agricultural soils [149]. MPs interact with heavy metals and metalloids such as arsenic, chromium, copper, cadmium, and lead present in the soil, causing harmful effects on its structure and microbial activities and subsequently impacting plants and human health. Some studies have been conducted on the combined pollution of MPs and heavy metals in soil and their interaction mechanisms. Physico-chemical factors such as aggregation, adsorption-desorption, complexation, precipitation, and electrostatic attraction can directly or indirectly influence the transport of heavy metals. MPs on their surface can indeed transport heavy metals into the environment [150,151,152], and their interaction can alter the nutrient content and subsequently influence soil bacterial communities [153,154]. Dong et al. [153] observed that polystyrene adsorbs arsenic through hydrogen bonds with the carboxyl groups. Wang et al. [154] studied the possible joint toxicity between polyethylene (PE) MPs and Cd in experiments carried out in microcosms consisting of soil samples grown with lettuce. The results showed that 10% MPs caused a reduction in plant biomass, soil pH, and cation exchange capacity but increased soil organic carbon. Furthermore, the MPs presence increased both Cd bioavailability in soil and its concentrations and accumulations in plants. Results suggest that contamination by MPs and heavy metals may therefore increase the toxicity, adsorption, accumulation, and bioavailability of heavy metals by altering the properties of the soil microenvironment [155].

Little is known about the effects of MPs on agroecosystems, particularly on soil microbial communities. Recently, it has been found that MPs influence the evolution of microbial communities and increase gene exchange, including the antibiotic and metal-resistance genes [79]. In a study of MPs collected from cotton fields in Xinjiang, China, the bacterial communities developed on them were examined [156]. The surfaces of MPs were found to be colonized by bacterial communities significantly different in structure from those of the surrounding soil, plant junk, and macroplastics, particularly from some polyethylene-degrading taxa (Actinobacteria, Bacteroidetes, and Proteobacteria). Overall, the results imply that MPs constitute a distinct habitat for bacteria in agricultural soil [156]. Some studies have established that MPs, by modifying the physical and biochemical properties of the soil, can influence a plant’s higher functions [15,67] (Figure 3).

### 5.3. Microplastics in Plants

In both terrestrial and aquatic ecosystems, plants are important in ecological balance and play irreplaceable roles in carbon and oxygen cycles. In addition, plants with large and irregular leaves can capture atmospheric particulate matter and act as phytoremediators of various pollutants, especially in urban environments [158,159].

They are exposed to MPs transported by air through wind or precipitation [160] and to MPs present in water and soil. The study of the interaction between plants (especially those grown for food) and MPs is, however, recent, and their impacts on them are little known; therefore, they should be better investigated due the persistence and wide distribution of MPs in air, water, and soil [30,161]. In addition to the plant species and growth stage, numerous factors influence the MPs phytotoxicity, such as the exposure dose, size, shape, type, and particle surface charges. Some studies showed that several MPs adverse effects depend on their size and dose [162,163]. Smaller particles have a higher bioavailability. Higher concentrations of MPs increase their interactions with plants. Due to their high adsorption capacity, MPs easily adhere to the surface of seeds, roots, and leaves, thus influencing the development and vital functions of plants, such as growth inhibition and seed germination. Some studies confirmed that roots can absorb MPs and transport them to the aerial parts (stem, leaf, flower, fruit, and seed). The leaves, on the other hand, can absorb MPs and subsequently transport them down to the roots [164]. According to Chen et al.’s 2022 study, MPs can be absorbed by plant roots and then enter and translocate into other tissues by transpiration [15]. The MPs and plants interaction not only causes them oxidative stress but also induces negative impacts on photosynthesis, metabolism, gene expression, and other growth parameters. Furthermore, the combination of MPs with other pollutants causes a more complex joint effect. The MPs phytotoxicity depends on both plastic particle characteristics, such as exposure dose, size, shape, type, age, and surface charge, and plant characteristics, such as species, tissues, and growth stage. The surface charge of MPs affects their interaction with plants. Negatively charged MPs inhibit plant growth more than positively charged ones [165]. Furthermore, it is more difficult for fibrous MPs to enter plants and penetrate tissues, but they are generally more toxic to organisms [50,166]. Irregularly shaped particles enter and transfer into plant tissues and, generally, are more toxic [62,65]. Cellular pressure, responsible for the plastic particles’ migration into plant tissue, can deform MPs in the intercellular space [167]; on the other hand, MPs can deform cell walls to enter into plant cells [168]. The size of the MP is an important factor affecting their uptake and translocation. Dong et al. [168], in a study carried out on carrots, saw that MPs of 1 μm can only enter the roots and accumulate at the intercellular level, while those of 0.2 μm size can migrate towards the leaves. MPs adsorbed by the roots could be translocated to other parts of the plant [119,169] and reaccumulated in the roots [170,171]. Liu et al. [172] stated that leaves of terrestrial plants can retain airborne MPs, thus acting as temporary storage. Earthworms can assimilate MPs present on fallen leaves and accumulate them near the roots. Contaminants present in the soil, such as bacteria, heavy metals, etc., could be adsorbed on MP particles and cause adverse effects on plants and microorganisms (Table 3) [160,173,174]. MPs can increase their phytotoxicity by adsorbing heavy metals. However, current knowledge of the coupled effect of their bioaccumulation is limited. It is hypothesized that they have a cumulative impact on chlorophyll content, photosynthetic activity, and induction of reactive oxygen species [175].

Microplastics can alter soil structure and composition, triggering a cascade of changes in the biophysical environment of the soil. The microplastic particles that caused the most noticeable effects differed substantially in shape, size, or composition from natural soil particles. Plants, in turn, adapt their traits to the new conditions [61].

Wang et al. [127] documented the biological effects of MPs on plants and explored the mechanisms. Due to their small size and high adsorption capacity, MPs can adhere to the seed and root surfaces, thus inhibiting seed germination and root elongation. Furthermore, they can interfere with water and nutrient uptake and ultimately repress plant growth. MPs can induce oxidative stress, cytotoxicity, and genotoxicity in plants, leading to many changes in their development, mineral nutrition mechanisms, photosynthesis, and accumulation of toxicants and metabolites in tissues. Crop productivity, food safety, and quality can be affected by MPs accumulation and subsequent plant damage, causing, at last, potential risks to human and animal health [127]. Plants can absorb MPs carrying on soil contaminants that would be transferred along the food chain. Studying the absorption of MPs by plants grown for human and animal food is therefore very important to establish whether they can enter plant cells because they are ubiquitous in soil and wastewater used for agricultural irrigation, and if accumulated in edible tissues, can affect food safety and pose risks to human health. Rillig et al. [180] were the first to discuss the potential impacts of MPs on plants and food safety. Several studies have confirmed that plants can absorb MPs by roots and translocate them out of the soil into aerial tissues [169,171,181,182,183]. These absorptions and those due to the foliar uptake [164,184] imply that MPs may enter the food chain and pose health risks to humans and livestock [127] (Figure 4).

Table 4 summarizes the significant effects observed in different crops exposed to MPs-contaminated soils. In general, the commonly measured agronomic parameters, such as plant height, fresh and dry aerial and root biomass, root length and diameter, and seed germination rate, show a tendency to decrease under MPs exposure; however, sometimes, an increase was found. These data must be treated with caution because, within the same crop, a given parameter may present notable differences due to various factors such as type of cultivar, measurement date, composition, size, dose, MPs aging period, location, experimental design, and the other cofactors’ presence.

## 6. Effects of Microplastics Exposure on Consumers and Farmers

Many studies have been conducted to explore the presence and effects of microplastics in soils and agricultural crops, but little is known about the effects of microplastics present in agriculture on farmers health. There are three main entry routes of microplastics and nanoplastics into the human body: ingestion, inhalation, and dermal contact, and farmers are exposed, in particular, to plastic microparticles through inhalation and dermal pathways. Only several studies have evaluated microplastic deposition in the human respiratory system, i.e., in the human lung and in the sputum [213,214,215,216,217] and effects on the cutaneous system [217,218,219,220]. Mainly studies on biological effects, factors influencing toxicity, and the probable mechanisms of cytotoxicity of MPs/NPs were conducted in vitro, using cell lines, in vivo, using aquatic organisms, and few studies used rats and mice as animal models. Keeping in mind that the effects of MPs/NPS are linked to size, charge, shape, and the presence of toxic additives or pollutants, and that the concentrations of MPs/NPs used in these studies were not equivalent to the concentration of MPs/NPs discovered in human tissues and fluids, many studies are still necessary to evaluate and quantify the risk of exposure to MPs/NPs for farmers and to minimize the release of NPs/MPs into the environment, developing effective mitigation strategies and establishing guidelines for proper management to reduce human uptake and to promote public health.

### 6.1. Exposure Routes

While the presence of NPs/MPs in food, water, and air has led to the assessment of their potential adverse effects on the population, their effects on the health of workers have been less studied. In the agricultural sector, the main routes of exposure to NPs/MPs are skin contact and inhalation of airborne particles, and not enough studies have been conducted to determine what happens to micro- and nanoplastic particles once they come into contact with the skin and respiratory tract.

In the epithelial barriers of the skin and respiratory tract, cell membranes exert a pivotal role in the maintenance of intracellular homeostasis, selective substance exchange, and signal transduction. MPs and NPs could interact with cell membranes, mainly composed of phospholipids and proteins, and hydrophobic actions are considered to be a common way for cell membrane absorption of NPs/MPs before they enter the cells [221]. Even at moderate levels, MPs/NPs could destroy important cell surface structures such as proteoglycans and other extracellular matrix components or hinder cell signaling processes that require interactions with extracellular ligands and cell surface receptors. Damaged cells released damage-associated molecular patterns (DAMPs), inducing the production of pro-inflammatory cytokines [222]. The surfactant molecules that are associated with most MPs/NPs preparations could be responsible for damage to the lipid bilayer of the plasma membrane, inducing cell death by necrosis [223]. Interactions of MPs/NPs with the plasma membrane and with endolysosomes would trigger cellular stress responses with the production of ROS. High levels of intracellular ROS can increase mitochondrial Ca^2+^, mitochondrial membrane depolarization, reduction of ATP, activation of cell death pathways, and release of pro-apoptotic factors leading to apoptosis. All these processes are intricately interconnected, therefore, induction of one process can lead to activation of other processes.

### 6.2. Skin Exposure and Health Effects

The skin is a natural barrier, which impedes NP/MP penetration. Thus, the skin pathway’s role in MPs diffusion has not been exhaustively explored. Although it is important to contemplate the MPs uptake through open wounds, sweat glands, and numerous hair follicles within the skin. In hair samples, high quantities of MPs were detected, and decreasing quantities were detected in facial and hand skin samples and saliva, respectively. MPs concentrations in scalp hair may be influenced by climatic conditions; in fact, in areas with higher humidity, which enhances MPs adherence to human receptors like hair as well, a higher number of particles were present [224]. Mechanical stress can facilitate the penetration of particles measuring 0.5 and 1 μm, allowing them to reach the epidermis and the dermis, and negatively charged particles of 0.5 μm can penetrate human skin, weakening the cytoskeleton and cell–matrix adhesion, increasing inflammatory responses and apoptosis, as reported by several studies confirming that the accumulation of MPs in the skin can lead to adverse effects [225,226,227]. Despite that, our comprehension of the potential harm and the repercussions of dermal exposure to MPs/NPs remains limited.

Uptake of particles across the skin requires penetration of the stratum corneum, which is limited to NPs particles smaller than 100 nm [228,229], while NPs of up to 200 nm could enter through the skin’s furrows, lipid channels, and vellus hair follicles. NPs can also accumulate on viable epidermis immediately below the stratum corneum and even intracellularly [230]. Accumulation of MPs/NPs was observed during zebrafish embryogenesis before the mouth was developed, demonstrating passive diffusion through the skin. Experimental data in zebrafish have shown that NPs can induce apoptosis of up to 54% of cell populations via skin diffusion [231]. Ku et al. demonstrated that chemical enhancers like oleic acid and ethanol could enhance the transdermal transport of nanoparticles by amplifying the intercellular lipid fluidity or removing lipids from the stratum corneum [232].

On keratinocytes, Langerhans cells, dendritic cells, melanocytes, macrophages, and T cells are present pathogen-associated molecular patterns and damage-associated molecular patterns receptors (PRR), which recognize MPs/NPs and lead to an immune response characterized by the secretion of antimicrobial peptides and pro-inflammatory molecules like interleukins (IL)–1, 6, 10, 18, 17, 22, tumor necrosis factor (TNF), and the recruitment of other immune cells. This may lead to an increased expression of proapoptotic proteins, such as BAX, caspase-3, caspase-8, caspase-9, and DR5, and activation of the mitochondrial apoptotic pathway, contributing to the overall impact of NPs/MPs on skin integrity [233]. Even though environmental factors are the main relevant causes of skin cancer, to date the effect of microplastics on skin cancer is not yet known. The study of Wang et al. showed that microplastics can promote tumor cell proliferation but also cause damage to normal skin [234]. The authors reported that microplastics were internalized into the skin squamous cell carcinoma cell line in a time- and dose-dependent manner, promoting the increase of mitochondrial ROS, which in turn caused a change in mitochondrial membrane potential, the opening of the mitochondrial permeability transition pore (mPTP), which in turn caused the release of mt-DNA from mitochondria into the cytoplasm, thus activating the inflammasome NLRP3 and ultimately causing skin cancer cell proliferation. Importantly, microplastics can cause damage to normal skin cells through NLRP3-mediated inflammation.

### 6.3. Inhalation and Health Effects

Present in the lung epithelial barrier are different epithelial cell types, such as goblet cells, club cells, ciliated cells, and basal cells in the airways and alveolar epithelial type I and II cells in the alveoli. From the upper airways to deep in the alveoli, the integrity of the lung barrier is crucial for respiratory health, protecting the lung from pathogens and environmental factors.

Inhaled plastic particles with a size of >1 μm may be eliminated by mucociliary clearance mechanisms, while larger MPs may translocate across the pleura and get absorbed by lung epithelial cells and accumulate in the respiratory tracts, causing acute and chronic respiratory difficulties, such as irritation of the respiratory tract, dyspnea, decreased lung capacity, coughing, increased phlegm production, interstitial fibrosis, and granulomatous lesions [235]. In vivo studies showed that rat and mouse alveolar macrophages of the respiratory tracts are able to phagocytose polystyrene particles in size-dependent manners, inducing inflammatory responses [236]. The small size of microplastics consents them to diffuse deep into the lungs, up to the alveoli, altering gas exchange with long-term effects, such as the progress of respiratory diseases and the potential diffusion to other organs. MPs/NPs can cross the alveolar barrier, pervade it, and, through the capillary blood system, disperse through the entire human body. Factors such as hydrophobicity, surface charge, surface functionalization, surrounding protein coronas, and particle size can influence absorption of microplastics in the lungs [237,238,239]. Nanoplastics transferred to the alveolar air–water interface cause dysfunction of pulmonary surfactant, leading to collapse of the pulmonary surfactant film, reducing the removal of particulates and pathogens inhaled from the alveoli and distal airways. In addition, exposure to microplastics caused a significant reduction of lung epithelial cell proliferation, induced epithelial cell apoptosis by upregulation of pro-apoptotic proteins, decreased oxidative stress-related GSH-Px, CAT, and SOD activities, induced ROS production, and caused redox imbalance. Extremely high concentrations of nanoparticles are cytotoxic for human bronchial epithelial cells and induce endoplasmic reticulum stress-related metabolic alterations [240]. Several in vitro studies reported that NPs induced disruption of epithelial barrier, reducing trans-epithelial resistance by depleting tight junction proteins and increasing expression of cellular matrix metallopeptidase 9 and surface-active protein A [241]. Lu et al. showed that microplastics administered via nasal drops are present in the airways, alveoli, and interstitium, demonstrating that microplastics can penetrate the alveolar epithelial barrier [242]. If inhaled f MPs are not eliminated by mucociliary clearance, in the lungs they trigger a localized biological response, with an overproduction of inflammatory mediators such as TGF-β, IL-6, IL-1β, and TNF-α [243,244]. By bronchoalveolar lavage fluid technique, the presence of MPs in lower airways and in lung tissue of cancer patients and healthy subjects was found, and polypropylene, polyethylene, and polycarbonate-based fibers can persist in extra-cellular lung fluid for about 6 months [215,245,246]. MPs/NPs diminish the tight junction proteins, decrease trans-epithelial resistance, and increase the expression of cellular matrix metallopeptidase 9 (MMP9), leading to lung injury and lowering the ability for lung repair, contributing to the pathogenesis of several lung diseases, such as asthma, COPD, and acute respiratory distress syndrome.

To date, limited information is available on the potential relation between MPs/NPs exposure and the development and progression of COPD; in fact, levels of MNPs in the lung tissue of patients with *chronic obstructive pulmonary disease* (COPD), with respect to healthy controls, have not been evaluated. The exposure of bronchial epithelial cell lines to polystyrene microparticles decreases the expression of α1-antitrypsin, increasing the risk for COPD development, and a comparable effect was observed after introducing polystyrene nanoplastics in a lung-on-a-chip model [247,248].

Studies on occupational and animal exposure suggested that MPs/NPs or constituents released from MPs/NPs can contribute to allergic asthma-related symptoms. The damage of alveolar barrier function increases the permeability of allergens, and the amount of MPs in the nasal lavage of patients with allergic rhinitis was found to be significantly higher than in the control group [249]. Co-exposure to NPs/MPs and di-(2-ethylhexyl) phthalate (DEHP) worsens allergic asthma by intensifying oxidative stress and inflammatory responses [250].

In summary, MPs/NPs possess the ability to impact the respiratory system, as evidenced in animal studies, although it is not completely established to what extent they contribute to the development of respiratory diseases. The most relevant dangerous effect of microplastics is the ability to act as a “Trojan Horse”, carrying toxic substances and pathogens into subjects who inhale, absorb, and ingest them [18].

### 6.4. Cellular Uptake

The cellular uptake of MPs and NPs depends on several factors, such as the size and the cell types, and MPs/NPs may enter the cell by disrupting cell membrane integrity or without affecting cell membrane integrity. The MPs–NPs interact and adhere to the cellular bilipid membrane by hydrophobic and van der Waals interactions, with electrostatic forces, as observed by Zhang et al. on basophilic leukemia (RBL-2H3) cells [251]. Although NPs/MPs may passively diffuse through the cell membrane, which is referred to as an adhesive interaction, or may be engaged by a channel protein or transport protein, they were typically uptaken by endocytosis mechanisms, such as phagocytosis, macropinocytosis, as well as clathrin- and caveolae-mediated endocytosis and clathrin- and caveolae-independent endocytosis (Figure 5).

The passive diffusion of NPs depends on the surface properties of the cell membrane, such as saturation kinetics, tension modulus, and degree of bending [252]. The endocytosis pathways of nanoplastics and microplastics depend on particle size and cell type, and in several situations, the combination of different endocytosis pathways contributed to the MPs and NPs uptake [253]. The upper size limit suggested for clathrin-mediated endocytosis was approximately 200 nm, while caveolin-mediated endocytosis is the mechanism for the internalization of particles larger than 500 nm. NP uptake was mainly by clathrin-dependent endocytosis, and clathrins-covered NPs could cross cell membranes, further releasing them into the cytoplasm by the formation of vesicles or alternatively could be internalized by the formation of caveolin, which induces the invagination of the cell membrane and maintains the stabilization of the structure [254]. During micropinocytosis, NPs could also be enclosed in the formed vesicles, which consisted of the large folds of cell membrane and extracellular fluid [255]. Microplastics > 1 μm can be engulfed by macrophages and neutrophils via phagocytosis/macropinocytosis [256].

MPs and NPs cell uptake by clathrin-dependent, caveolin-dependent, phagocytosis, pinocytosis, or injury of cellular membrane causes reactive oxygen species (ROS) generation from the mitochondria, an increase in mitochondrial Ca^2+^, mitochondrial membrane depolarization (↓ΔΨm), increased pro-apoptotic factors, reduction in ATP, and the following production/release of pro-inflammatory cytokines.

### 6.5. Potential Mechanisms of Cellular Toxicity Induced by Micro/Nanoplastics

The damage of cell membranes, induction of oxidative stress, and immune response are the most probable mechanisms of MP/NP toxicity [222]. Holloczki et al. reported that NPs were found to penetrate into the hydrophobic milieu of the plasma membrane bilayer and bring about structural changes [257].

The MPs/NPs that are endocytosed were encapsuled in the endosome that can be transferred from cell membranes to lysosomes containing hydrolytic enzymes, can induce the permeabilization of the lysosomal membrane the release of lytic enzymes into the cytosol, which in turn would interact and damage the other intracellular organelles [258,259,260] such as mitochondria, causing pore formation in the plasma membrane and subsequent reactive oxygen species (ROS) generation [261,262] or interact with the nucleus interfering with mitotic spindle formation and chromosome migration during cell division, or interact with the endoplasmic reticulum inducing stress with accumulation of misfolded proteins. MPs/NPs genotoxicity may be facilitated by direct or indirect DNA impairment through particle or ROS translocation into the nucleus and injury of DNA replication/repair machinery.

Increased levels of intracellular ROS can induce an increase in mitochondrial Ca^2+^, mitochondrial membrane depolarization (↓ΔΨm), reduction in ATP, release of pro-apoptotic factors Bax and cytochrome C, expression of autophagy-related genes, and damage-associated molecular patterns (DAMPs) from mitochondria, resulting in production/release of pro-inflammatory cytokines and cell death (Fig. X) [260,263,264]. Some studies have shown that single-dose and chronic long-term MPs/NPs exposure causes oxidative stress-mediated restriction of cell growth, generating autophagic structures, and premature aging [265]. Using both human and animal models, several studies have pointed out that even at low concentrations, plastic particles can induce cellular oxidative stress, and extreme quantities lead to cytotoxicity [266,267]. A study in rotifers has highlighted that MPs increase the levels of ROS in a size-dependent way, leading to the activation of antioxidant-related enzymes, including superoxide dismutase, glutathione s-transferases, glutathione reductase, and glutathione peroxidase, aimed at mitigating the oxidative stress caused by microplastic exposure [268].

MPs/NPs interact with proteins, membrane transporters, and cellular receptors and may cause changes in protein folding; some amino acids with nonpolar side chains interact with NPs and modify the protein’s secondary structure, leading to the production of dysfunctional proteins [269,270]. Furthermore, proteins can bind to plastic particles and form “protein coronas” around them. The composition of a protein corona is important for the biological response to protein-coated MPs/NPs that may be different if compared to that of bare particles, consequently influencing the interaction and absorption of MPs/NPs. Some serum proteins contribute to decreased NPs endocytosis, while others may enhance the NPs uptake or modulate the endocytic mechanisms [271,272].

The contact between cells and NPs–MPs induces the production of inflammatory cytokines such as TNF-a, IL-1b, and IFN-g, leading to local and systemic inflammation by activating nuclear factor kappa B (NF-kB) and cGas/STING signaling pathways, which could be related to the increased cytotoxicity, as observed in studies performed on different cell lines [247,273,274]. Toxicity of MPs/NPs is very complex and can be due to intrinsic toxicity of the particles based on polymer type, size, charge, morphology, and biopersistence leading to bioaccumulation, but also to the presence of toxic additives or to the ability to act as a carrier for pathogens and toxic environmental pollutants, but also to dose and exposure time that spread MPs/NPs internalization.

To date, few studies have provided information concerning the exposure of workers, the distribution of MPs within the body tissues, and the potential cocktail effect MPs might exert with other concurrent contaminants. Described effects in the respiratory tract include epithelial barrier dysfunction, cytotoxicity, inflammatory response, and redox imbalance and synergistic effects with allergens. Effects of microplastics on skin include damage to normal skin, irritation, or promotion of tumor cell proliferation. Human exposure through skin contact or inhalation with microplastics containing toxic chemicals contributes to the absorption of these substances, increasing their body burden (Figure 6).

Major toxic effects of MPs and NPs are cell viability, oxidative stress, inflammation, cytotoxicity, DNA damage, disruption of metabolism, and tumor cell proliferation. More detailed and complete studies of MPs and NPs on health risks for farmers from inhalation and dermal contact exposure are needed.

## 7. Possible Alternatives to the Use of Plastic in Agriculture

The use of biodegradable plastics in agriculture could be an alternative to minimize MPs pollution even if the effects of biodegradable plastic on soils are not fully known.

Recently, the development and use of biodegradable packaging materials for new food packaging have assumed an increasing role in agriculture and food processing. This kind of packaging is investigated for food preservation and shelf-life extension and is replacing synthetic plastic use. Biopolymer-derived biofilms, such as starch, cellulose, chitosan, pectin, agar, alginate, etc., reinforced through different nanofillers, such as mineral clays and fibers (zeolite, kaolin, montmorillonite, vermiculite, lignin, etc.), have been used to replace synthetic plastics in packaging for agricultural technical means, feed, and food. El Bourakadi et al. described the active biofilms based on biodegradable polymers and the biological and physical effects of several nanofillers used, by discussing their role in food preservation and engineering [275]. Malinconico, in 2017, clarified the materials design, development, and utilization of biodegradable, soil-compostable bioplastics for agriculture. This author discussed technologies, developments, and applications available in the “plasticulture” area, including mulching, covers, tunnels, silage bags, pheromone traps, fertilizers, pesticides, hormones and seeds coating, nursery pots, and containers. The results of field experiences of biodegradable plastics use and their effect on soil protection, crop production, and post-harvest management were presented [276]. Other authors [277] demonstrated that soil mulching conserves water, reducing the drought effects, heat stress, and the unwise use of limited water during the cropping season by minimizing evaporation. Improving soil moisture conservation is essential in the case of restricted and controlled water resources. In this study, the effects of plastic film, crop straw, gravel, volcanic ash, rock pieces, sand, concrete, paper pellets, and livestock manures were compared. Mulching reduced soil water loss and soil erosion, enriched soil fauna, and improved soil properties and nutrient cycling in the soil. Furthermore, mulching reduced runoff and soil loss, and it reduced water evaporation, reducing the crop irrigation demand. This study demonstrated that it is very important for farmers to choose natural mulching rather than synthetic applications [277]. As well as the potential soil moisture conservation, mulching suppresses the weed population. Different green mulching showed significant positive effects on the growth, yield, and quality of various crops. Green mulches are an available and cheap source to control weed populations and to preserve the soil moisture contents. Hence, the use of green soil covering could partially balance the water requirement of crops in limited water conditions. Likewise, integrating the mulching use, in the case of wheat straw, cotton sticks, black plastic, and maize straw, with partial root-zone drying could complete this technique to enhance overall crop growth, development, and yield [278]. Concerning the replacement of synthetic fertilizers that could contain MPs, compost can totally or partially substitute mineral nitrogen fertilization, as it emerged in a three-year herbaceous crop succession in a field trial [279]. Four types of compost and eight fertilization treatments were evaluated. The compost application did not show a significant effect on soil total organic carbon. Effects on crop yield have shown no substantial difference between compost and synthetic fertilizers. Compost fertilization did not generate an increase of NO_3_-N leaching in the percolation water. Obtained results showed the good fertilization properties of compost compared to synthetic fertilizers, while the amendment effect was not relevant, probably due to the low doses applied and the short trial period [279].

## 8. Conclusions

Found in air, water, and soil, MPs can persist in the environment, enter the food chain, and accumulate in living organisms. They can also be substrates for biological and chemical environmental contaminants, acting as potential vectors for their long-range transport.

Today, in Europe, there is an ongoing debate on possible strategies to reduce the use of plastic and make it reusable and recyclable. These measures include the Single-Use Plastics Directive and the proposal for a new EU packaging regulation [280]. Both aim to reduce plastic waste generation by encouraging container recycling and the alternative use of eco-sustainable materials [280]. Also, the European Commission has adopted measures under the REACH regulation to limit the intentional addition of microplastics to products (agricultural fertilizers, pesticides, etc.). Plastic is widely used in agriculture (seed containers, mulch films, irrigation pipes, bags, and containers of plant protection products, greenhouse films, fruit collection boxes, and fertilizer storage bags). MPs resulting from their disintegration can alter the physical structure of the soil, be absorbed by plants, and interfere with root growth and nutrient uptake. From edible plants through the food chain, they can reach humans and significantly impact their health. Appropriate studies are needed to better understand the mechanisms by which edible plants trap and accumulate MPs in their tissues, their transfer to different trophic levels of terrestrial food chains, and the associated risks to agricultural sustainability, nutrition, and human and animal safety [201]. The effects of acute exposure to high concentrations of microplastics, chronic exposure, and possible long-term effects due to the persistence of MPs in the environment have not been adequately studied, and it is not yet known how harmful they may be to the environment and human safety [176]. In particular, the effects of MPs on agricultural workers are not yet known, so the occupational risk of exposure to MPs in agriculture cannot yet be assessed. It would be important to study this aspect by highlighting that agricultural workers are potentially at risk of occupational exposure to MPs, both from their presence in the environment and from the plastic materials used in agricultural degradation. New studies are needed to determine if and to what extent MPs are harmful to agricultural workers, to decide whether it is necessary to use protective devices, and to propose alternative materials to plastic in agriculture to protect their health. In addition, it would be appropriate to use alternative agronomic practices, such as those used in organic and synergic agriculture under the European Commission’s regulations to promote sustainable agriculture.

## Figures and Tables

**Figure 1 ijerph-22-00045-f001:**
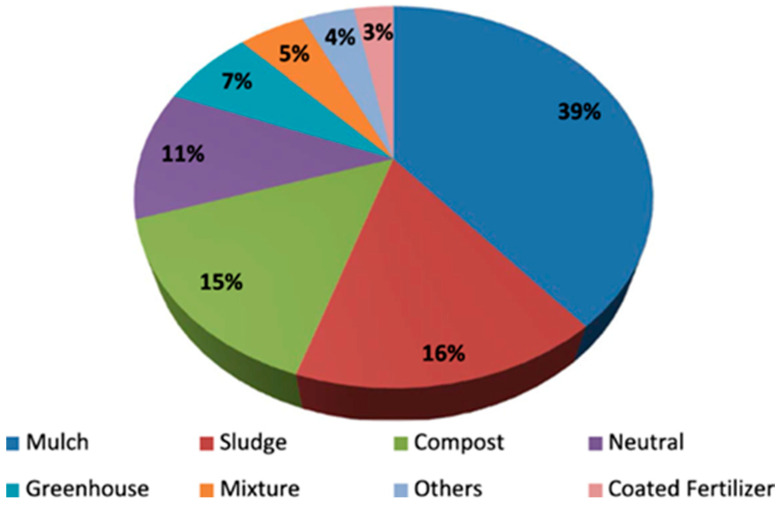
Microplastics sources in agricultural soils (*n* = 120) [77].

**Figure 2 ijerph-22-00045-f002:**
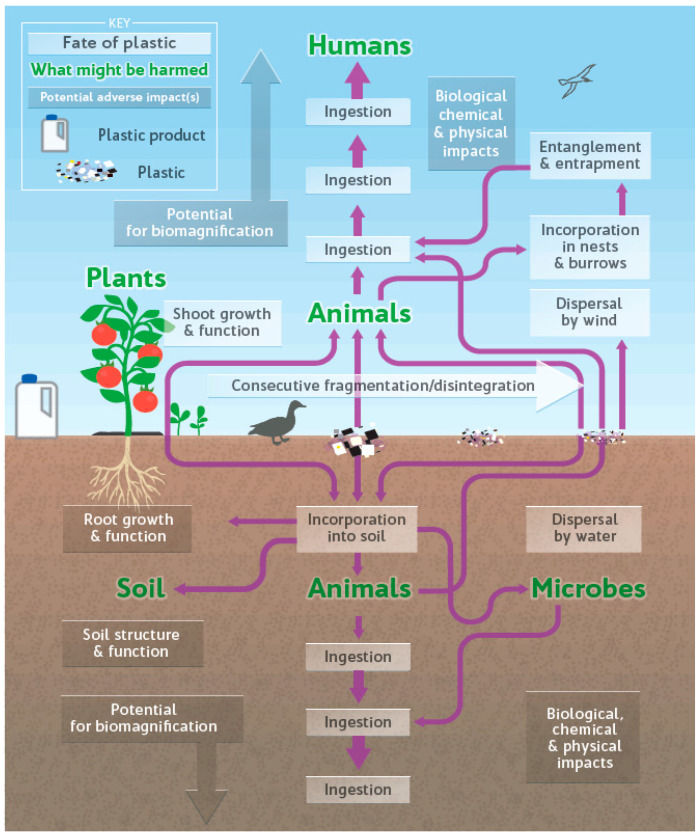
Schematic representation of the flow of plastic in terrestrial environments. How plastic enters into the Earth’s ecosystems and into our food chain [87].

**Figure 3 ijerph-22-00045-f003:**
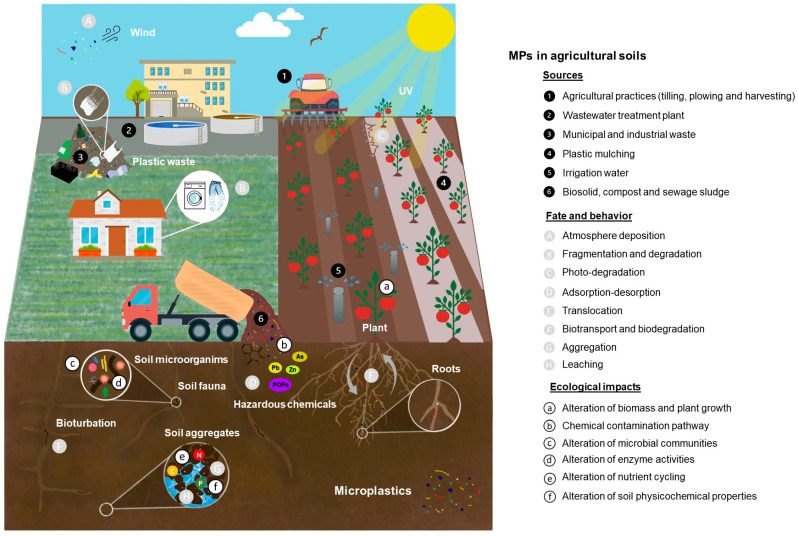
MPs in agricultural soil. Sources, fate, behavior, and ecological impacts [157].

**Figure 4 ijerph-22-00045-f004:**
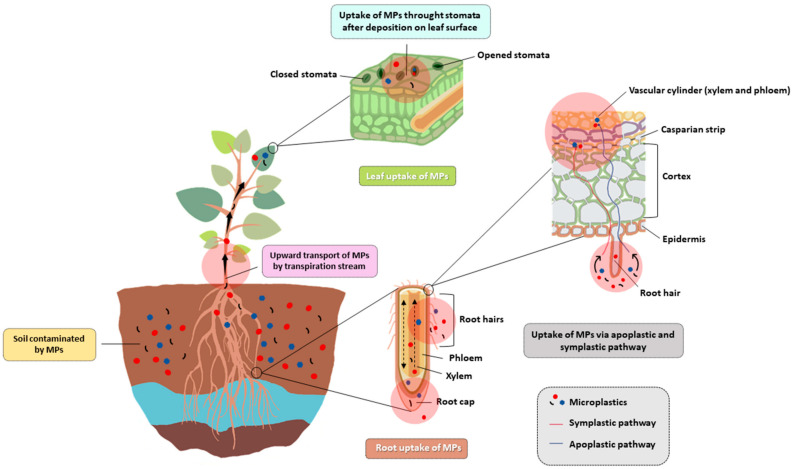
Uptake and translocation mechanisms of MPs in plants [157].

**Figure 5 ijerph-22-00045-f005:**
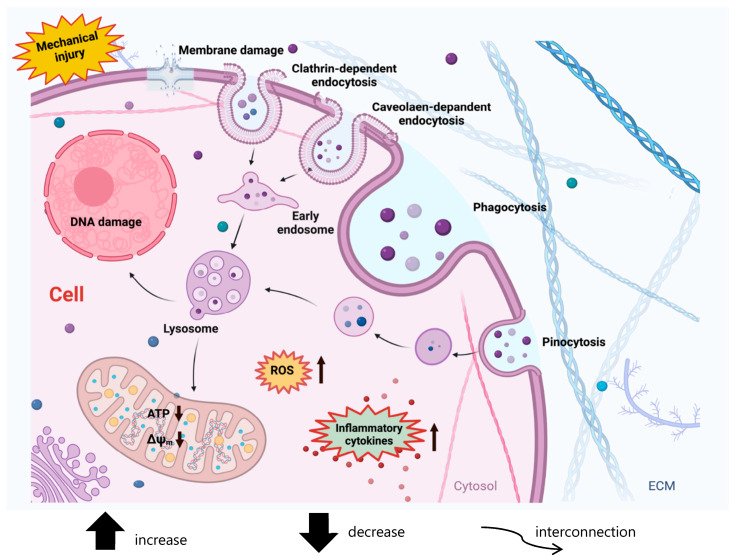
Cellular uptake and cytotoxic mechanisms of MPs/-NPLs. MPs and NPs can be intaken into cells by different mechanisms such as clathrin-dependent, caveolin- dependent, clathrin/caveolin-independent, phagocytosis and pinocytosis. The cytotoxic mechanisms of MPs and NPs included the membrane damage, oxidative stress and increases of ROS production, induction of inflammatory cytokines, DNA damage, disruption of mitochondrial energy homeostasis and metabolism, and all these mechanisms are closely interconnected with each other.

**Figure 6 ijerph-22-00045-f006:**
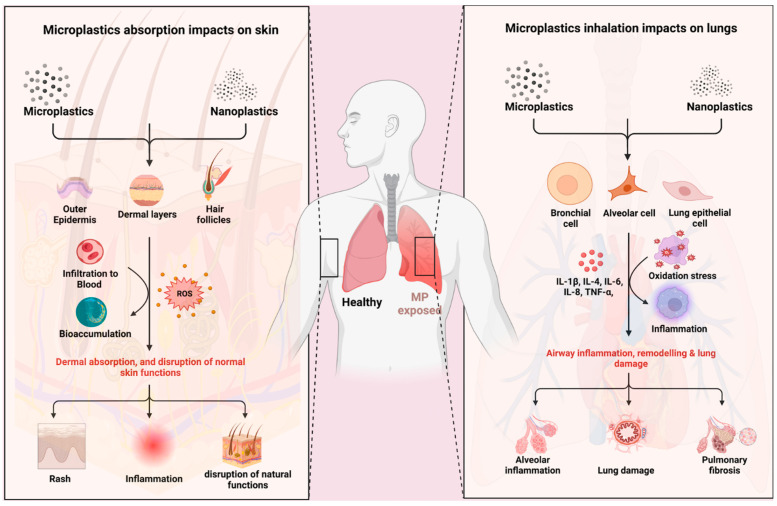
Effects of MPs/NPs absorbed by inhalation and dermal contact.

**Table 1 ijerph-22-00045-t001:** Most used plastic equipment in agriculture.

Most Used Plastic Equipment in Agriculture				
Crop Protection Films	Nets	Irrigation/Drainage Pipes	Packaging	Other
Greenhouses and tunnels	Anti-hail	Water collection	Fertilizer storage bags	Silage film
Mulching sheets	Anti-bird	Channel linings	Agrochemical containers	Fumigation film
Vineyard and orchard coverings	Windbreak	Irrigation tapes and pipes	Generic containers	Strings and wraps for round and square bales
Nursery films	For shading	Drainage pipes	Liquid storage tanks	Nursery pots
	For harvesting olives, almonds, walnuts, and hazelnuts	Micro-irrigation	Cages	Ropes
		Drippers	Fruit storage and protection	Filters, clips, and connectors for irrigation systems

**Table 2 ijerph-22-00045-t002:** Most used plastic in agriculture.

Most Used Plastic in Agriculture	
Plastic	Use
low-density polyethylene (LDPE), linear low-density polyethylene (LLDPE), and high-density polyethylene (HDPE)	greenhouse and mulch films
polyvinyl chloride (PVC)	drip irrigation pipes
polyethylene terephthalate (PET)	packaging and bottling
polypropylene (PP)	strings or ropes
polyurethane (PU-PUR)	protecting storage
polystyrene (PS), polycarbonate (PC), and polymethyl methacrylate (PMMA)	seedling and plant trays

**Table 3 ijerph-22-00045-t003:** Effects of the most commonly adsorbed heavy metals on MPs.

Effects of the Most Commonly Adsorbed Heavy Metals on MPs			
Heavy Metal	Plant	Effects	Bibliography
Cu, Pb (PE)	Rapeseed	Malondialdehyde content increase and deterioration of plants increase.	[176]
Cd (MPs)	Strawberry	Decrease in stem diameter (38%), plant growth (33.4%), and root growth (15%).	[177]
As (MPs) As (PMMA)	Rapeseed	Germination inhibition Reduction of germination index, biomass, root, and seed length.	[153]
As, Cu, Cd (MPs)	Rapeseed	Inhibition of respiration, photosynthesis, and plant growth.	[153]
Mn, As, Cd (PE)	Rice	Reduce Mn, As, and Cd bioaccumulation. Increase Na bioaccumulation.	[178]
Cu, Cd (MPs)	Wheat	Increase chlorophyll content and photosynthetic activity. Reduce the reactive oxygen species (ROS) accumulation	[179]
As (PS)	Carrot	Increase As absorption. Increase oxidative activities.	[168]

**Table 4 ijerph-22-00045-t004:** The significant effects observed in different crops exposed to MPs-contaminated soils. Bio: biodegradable: plastic; HDPE: high-density polyethylene; LDPE: low-density polyethylene; PBAT: poly-butylene-adipate-co-terephthalate; PC: polycarbonate; PE: polyethylene; PET: polyethylene: terephthalate; PHBV: poly: (3-hydroxybutyrate-co-3-hydroxyvalerate; PLA: polylactic acid; PMMA: polymethyl: methacrylate; PP: polypropylene; PS: polystyrene; PU: polyurethane; PVC: polyvinyl: chloride.

Plant	MPs	Effects	References
*Oryza sativa*	PS	Poor radicle development	[184]
	PMMA	Oxidative stress (increased H_2_O_2_ and O_2_^−^ in roots)	
		Changes in carbohydrate and ROS metabolisms in roots and leaves with different trend	
	PS	Biosynthesis of amino acids, nucleic acids, fatty acids, and some secondary metabolites decrease	[185]
		Biomass decrease	
*Triticum aestivum*	PS, PMMA	Inhibition of vegetative growth and reproduction	[186]
		Hydroponic growing with wastewater Crack-entry mode Transport of particles from roots to shoots	[171]
		Higher wicking rates increase the absorption of plastic particles. Transport through the xylem	[187]
	PS	Root surface, particularly root tip electrostatic forces: negatively charged particles repelled by root surface cells (which are also negatively charged).	[69]
		MPs are not absorbed by tissues. Accumulation of plastic particles around root cap cells and along the root surface. There is no evidence that plastic particles penetrate cellular structures or between cells. Adhesion of MPs to root cap cells is potentially dangerous for roots and edible tubers (carrots, potatoes, beets, etc.).	[188]
	PVC, PE	Both PVC and PE influenced the sprouting and development of roots and plant biomass in a dose ratio dependent on the amount of MPs present in the soil.	[189]
	PHBV	PHBV caused a decrease in seed germination and all treated plants died during the 4 weeks of the experiment. PHBV likely blocks the uptake of essential nutrients induced	[190]
*Allium fistulosum*	PE, PET, PP, PS	Plants and soil are highly susceptible to the MPs presence. The damage extent depends on the MP and varies with the size, shape, and chemical composition of the pollutant. Root and leaf different effect in compositions.	[61]
*Lolium perenne*	PS	Blocking pores in seed capsule and especially on the root hairs but also found on the leaves and epidermis	[191,192]
*Vicia faba*	PS	In root tip cells Genotoxic and oxidative stress were induced PS-MPs produced ROS and induced oxidative damage Root relative elongation and weight decreased	[66]
*Zea mays*	PE, PS, PLA	MPs cause a decline in transpiration, nitrogen content, and growth	[193,194]
	PE, PLA	MPs accumulate on the rhizosphere impair water and nutrient uptake Biomass Decreased in PLA	[195]
	PU	Cultivar ZTN 182 increased height and aerial dry biomass in 6-week-old plants exposed to 1% PU	[149]
*Lactuca sativa*	PS, PMMA	PS and PMMA penetrating the stele of wheat and lettuce using the crack-entry mode Uptake via cracks in the epidermis and features of the polymeric particles (density, hydrophobic surface, adhesive properties, and mechanical flexibility), translocation due to transpirational pull through the xylem enhanced by higher temperature and lower humidity. Particles could be transported from the roots to the shoots.	[169]
	PE	Higher transpiration rates enhanced the uptake of plastic particles. Weight decreased. Height decreased. Root length decreased. Leaf number decreased. Chlorophyll *a*/a+b decreased	[161]
	PVC-a (100 nm–18 μm) PVC-b (18–150 μm) (0.5%, 1%, e 2%)	No significant effect on root activity. PVC-a 0.5% and 1% increased total length, surface area, volume, and diameter of roots. In the leaves no significant effect on malondialdehyde content, but PVC-a 1% increased superoxide dismutase activity. Carotenoid synthesis was promoted by PVC-a but inhibited by PVC-b. PVC-a 1% could reduce the absorption, dissipation, capture, and transfer capacity of electrons of light energy.	[196]
*Brassica rapa*	PS	PS influenced the plant photosynthesis and growth parameters MPs affected the soil properties, rhizosphere microbial community composition The different sizes of PS-beads affected the different parameters	[197]
	PBAT	No effect on plant growth	[198]
*Brassica chinensis* L.	HDPE	Fresh weight reduction	[199]
	PS	Reduction of soluble sugar, starch, and chlorophyll in leaves	
*Brassica juncea*	HDPE	Lower concentrations of phenolic content, chlorophyll and proline Transfer of MPs through roots and leaves with potential damage to plants	[200]
*Phaseolus vulgaris*	LDPE, PLA mixed with PBAT	LDPE had limited effects on common bean growth Biomass: No significant effect. Leaf area increased PLA strongly reduced shoot, root biomass, and fruit biomass MPs increased specific root length/nodules strongly	[201]
*Glycine max*	PS	Significantly limits root development and length	[202,203,204]
*Hordeum vulgare*	PS	Significantly limits root development and length	[205,206,207]
*Lepidium sativum*	PP, PE, PVC, PE + PVC	Germination inhibited shoot height/lowered biomass/raised leaf percentage inhibition of seed germination, plant growth, and number of leaves, oxidative stress in roots	[50]
	PBAT	No effect on plant growth	[50]
	PET	Inhibition of germination rate, leaf number and fresh biomass. Reduced photosynthetic efficiency (chl-a/chl-b imbalance)	[15,208,209]
	PP, PET	Different sizes of MP affect differently the cress growth and development	[208]
	MPs and NPs	Plastic particles accumulating in the pores of *L. sativum* seed capsules could inhibit water absorption and retard germination and root growth	[162]
	PC	Aging polycarbonate in a thermal chamber (high temperatures) reduces the ecotoxicity of plastic particles, i.e., inhibition of seed germination	[210]
*Avena sativa*	PBAT	No effect on plant growth	[211]
*Rafanus sativus*	PBAT	No effect on plant growth	[211]
*Cucurbita pepo* L.	PP, PE, PVC and PET	All MPs impaired the growth of roots and, above all, shoots. Reduction of leaf size, chlorophyll content, and photosynthetic efficiency, changes in micro- and macro-element profiles. PVC decreased leaf blade size, photosynthetic performance index values, and iron concentration in the plant to a greater extent than other treatments.	[68]
*Solanum lycopersicum*	HDPE, LDPE, PP and PET	Soils with sludge containing MPs promoted the growth of tomato plants.	[212]

## Data Availability

All the reported data come from the publications cited in the bibliography.

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
