# Peer review of "Microplastics in Agricultural Crops and Their Possible Impact on Farmers’ Health: A Review"

_ijerph, 2024, doi:10.3390/ijerph22010045_

Round 1
Reviewer 1 Report
Comments and Suggestions for Authors
I am thankful for the opportunity to review this manuscript, a useful meticulous research paper that analyzes the exceptional informations of the impact of microplastics in agricultural crops on farmers health. However, the clarity with which they have structured the review does not allow readers to navigate through complex information with ease. The manuscript, that matched the area of interest of the journal, is interesting and would attract wide readership. In my opinion, is not well written. There are some critical issues that authors need to carefully consider to improve the scientific outlook and proficiency level of the paper.
The objectives are poorly written, please rewrite. The introduction needs a lot of work. There is no clear logical flow between sections.
There are very long sentences, such as from line 37 to 122, or from line 708 to 751. Please fragment.
In the title mention “Farmers Health”, only farmers?
Some words are underlined in yellow? Or that in line 716 the font is larger? Does this have any significance?
Why is page 6 empty?
Figures 3,4, 5,6,7 and 8 are practically invisible.
Table 4 as it appears is unacceptable.
The conclusions do not respond to the most conclusive thing, please rewrite.
Author Response
Comments 1: The objectives are poorly written, please rewrite. The introduction needs a lot of work. There is no clear logical flow between sections.
Introduction has been rewritten
Comments 2: There are very long sentences, such as from line 37 to 122, or from line 708 to 751. Please fragment.
Long sentences have been fragmented
Comments 3: In the title mention “Farmers Health”, only farmers?
The title of this paper mentions "farmer health" because the aim of this article is also to highlight that agricultural workers may be at risk from occupational exposure to MPs, although there are few studies on these expositions.
Comments 4: Some words are underlined in yellow? Or that in line 716 the font is larger? Does this have any significance?
There were some typos that has been corrected.
Comments 5: Why is page 6 empty?
The paper has been revised to resolve this typo.
Comments 6: Figures 3,4, 5,6,7 and 8 are practically invisible
Figures 3, 4, 5, 6, 7, and 8 have been resized to be readable.
Comments 7: Table 4 as it appears is unacceptable.
Table 4 has been rewritten.
Comments 8: The conclusions do not respond to the most conclusive thing, please rewrite.
This article aims to highlight that agricultural workers are potentially at risk of occupational exposure to MPs from both environmental contamination and degradation of plastic materials used in agriculture, although there are only few studies about this risk.
Reviewer 2 Report
Comments and Suggestions for Authors
This paper reviews the nature of microplastics, their distribution and effect in the atmosphere, soil and plants, ecotoxicity and human toxicity. The content involves many aspects of the environmental impact of microplastics, the content is relatively comprehensive. But more author's own point of view would be better. Specific comments are as follows:
1. The title of the article needs to be changed. The title points out the keyword Farmers Health, but the author describes the toxicity and health effects of microplastics as applying to all human beings, and doesn't show the particular health impacts of farmers as a special group.
2. The abstract needs to be revised to be less redundant and more informative, as well as conclusive, so that the reader has a clear idea of the main points of the manuscript.
3. The categorisation of the manuscript does not make sense.
First of all, the Introduction showcases too many research directions, and it would be more organised if it was divided into subheadings according to the content. Moreover, some conclusions are repeated from Introduction to “2 Microplastics in Agriculture”, and the repeated contents should be transferred out of the introduction and merged, such as the classification, properties, adsorption and toxicity of microplastics, etc.
Secondly, it is suggested that 2.1-2.3 on the distribution of microplastics should be a separate chapter, 2.4-2.9 on the health effects of microplastics should be a separate chapter, and 2.10 on Alternatives and related policies should be a separate chapter.
4. The section of discussion of research gaps or future research perspectives should be added.
5. Table 4 reformatted to take up fewer pages.
Author Response
Comments 1: The title of the article needs to be changed. The title points out the keyword Farmers Health, but the author describes the toxicity and health effects of microplastics as applying to all human beings, and doesn't show the particular health impacts of farmers as a special group.
The title of this paper mentions "farmer health" because the aim of this article is also to highlight that agricultural workers may be at risk from occupational exposure to MPs, despite the fact that there are few studies on these expositions.
Comments 2. The abstract needs to be revised to be less redundant and more informative, as well as conclusive, so that the reader has a clear idea of the main points of the manuscript.
The abstract has been reviewed.
Comments 3. The categorisation of the manuscript does not make sense.
First of all, the Introduction showcases too many research directions, and it would be more organised if it was divided into subheadings according to the content. Moreover, some conclusions are repeated from Introduction to “2 Microplastics in Agriculture”, and the repeated contents should be transferred out of the introduction and merged, such as the classification, properties, adsorption and toxicity of microplastics, etc.
Secondly, it is suggested that 2.1-2.3 on the distribution of microplastics should be a separate chapter, 2.4-2.9 on the health effects of microplastics should be a separate chapter, and 2.10 on Alternatives and related policies should be a separate chapter.
Sections of the document have been redefined.
Comments 4. The section of discussion of research gaps or future research perspectives should be added.
The discussion has been added at the end of the introduction and in the conclusions.
Comments 5. Table 4 reformatted to take up fewer pages.
Table 4 has been rewritten.
Round 2
Reviewer 1 Report
Comments and Suggestions for Authors
The authors have addressed all my suggestions, and the manuscript is now significantly improved and acceptable for publication.